# Post-Exercise Protein Intake May Reduce Time in Hypoglycemia Following Moderate-Intensity Continuous Exercise among Adults with Type 1 Diabetes

**DOI:** 10.3390/nu15194268

**Published:** 2023-10-06

**Authors:** Franklin R. Muntis, Elizabeth J. Mayer-Davis, Saame R. Shaikh, Jamie Crandell, Kelly R. Evenson, Abbie E. Smith-Ryan

**Affiliations:** 1Department of Nutrition, Gillings School of Global Public Health, University of North Carolina, Chapel Hill, NC 27599, USA; frmuntis@email.unc.edu (F.R.M.); shaikhsa@email.unc.edu (S.R.S.); 2Department of Medicine, University of North Carolina, Chapel Hill, NC 27599, USA; 3School of Nursing, University of North Carolina, Chapel Hill, NC 27599, USA; jbigelow@email.unc.edu; 4Department of Biostatistics, Gillings School of Global Public Health, University of North Carolina, Chapel Hill, NC 27599, USA; 5Department of Epidemiology, Gillings School of Global Public Health, University of North Carolina, Chapel Hill, NC 27599, USA; kelly_evenson@unc.edu; 6Department of Exercise & Sports Science, University of North Carolina, Chapel Hill, NC 27599, USA; abbsmith@email.unc.edu

**Keywords:** type 1 diabetes, exercise, protein, sports nutrition, glycemia, hypoglycemia, adults, continuous glucose monitoring, high intensity interval training, moderate intensity continuous training

## Abstract

Little is known about the role of post-exercise protein intake on post-exercise glycemia. Secondary analyses were conducted to evaluate the role of post-exercise protein intake on post-exercise glycemia using data from an exercise pilot study. Adults with T1D (*n* = 11), with an average age of 33.0 ± 11.4 years and BMI of 25.1 ± 3.4, participated in isoenergetic sessions of high-intensity interval training (HIIT) or moderate-intensity continuous training (MICT). Participants completed food records on the days of exercise and provided continuous glucose monitoring data throughout the study, from which time in range (TIR, 70–180 mg/dL), time above range (TAR, >180 mg/dL), and time below range (TBR, <70 mg/dL) were calculated from exercise cessation until the following morning. Mixed effects regression models, adjusted for carbohydrate intake, diabetes duration, and lean mass, assessed the relationship between post-exercise protein intake on TIR, TAR, and TBR following exercise. No association was observed between protein intake and TIR, TAR, or TBR (*p*-values ≥ 0.07); however, a borderline significant reduction of −1.9% (95% CI: −3.9%, 0.0%; *p* = 0.05) TBR per 20 g protein was observed following MICT in analyses stratified by exercise mode. Increasing post-exercise protein intake may be a promising strategy to mitigate the risk of hypoglycemia following MICT.

## 1. Introduction

For those living with T1D, the risk of developing cardiovascular disease is estimated to be ten times greater than those without diabetes and is the leading cause of morbidity and mortality [1]. Intensive insulin therapy has been shown to reduce the risk of cardiovascular disease events and mortality among people with T1D by 57% and 42%, respectively; however, it has also been shown to promote weight gain [2]. Among those who gain excessive weight, the benefits of intensive insulin therapy are substantially reduced [3]. Regular exercise is central to both diabetes and weight management and promotes reductions in body mass and BMI and improvements in glycemia and cardiorespiratory fitness; however, as many as 60% of adults with T1D report not participating in any regular weekly exercise [4]. For people living with T1D, exercise is associated with an increased risk of hypoglycemia during and at least 24 h following exercise, and the fear of hypoglycemia is commonly cited as a leading barrier to participating in regular physical activity [5,6].

Nutrition plays an important role in managing glycemia, and expert consensus on carbohydrate strategies to promote improved glycemia during and following exercise has been published [7]. However, much less is known about the role of protein intake in the management of exercise-related glycemia. For the general population, the American Dietetic Association recommends consuming 0.25–0.3 g/kg of dietary protein every 3–4 h following exercise to support recovery from and adaption to regular exercise training [8]. Among healthy adults, increasing dietary protein intake promotes numerous physiological benefits when paired with regular exercise, including improved recovery times, reduced muscular soreness, improvements in muscular strength and hypertrophy, and, when paired with a hypocaloric diet, greater reductions in fat mass and greater retention of lean mass [9,10,11,12,13,14]. Among people living with T1D, protein has also been shown to cause a dose-dependent elevation in blood glucose levels, peaking around 3 h after consumption and remaining elevated for at least 5 h and possibly as long as 12 h [15,16,17,18,19,20]. In theory, increasing dietary protein intake following exercise among people with T1D may mitigate declining glycemia following exercise, thereby reducing the risk of hypoglycemia. 

Only two studies, to the authors’ knowledge, have examined the effects of peri-exercise protein intake on exercise-related glycemia among people with T1D [21]. A small (*n* = 10) randomized trial of adolescents with T1D measured glycemia during moderate-intensity continuous cycling exercise performed following different nutritional strategies found that a protein-supplemented breakfast consumed 2 h prior to exercise was effective at preventing hypoglycemia during exercise compared to a standard breakfast [21]. Similarly, in a laboratory-based pilot study of six participants with T1D with a mean age of 20.2 ± 3.1 years and an average BMI of 26.7 ± 5.0, a 50 g protein bolus provided 3.25 h after 45 min of moderate-intensity exercise significantly reduced the glucose requirements required to maintain euglycemia overnight compared to water, indicating that post-exercise protein intake may reduce the risk of late-onset post-exercise hypoglycemia [22]. These studies, however, were conducted in well-controlled environments and may not fully represent the effects of protein intake in a free-living environment. 

The primary aim of this study was to conduct secondary analyses of data from an acute exercise pilot study among adults with T1D to examine the relationship between post-exercise protein intake and glycemia following isoenergetic bouts of high-intensity interval training (HIIT) or moderate-intensity continuous training (MICT). We hypothesized that elevating protein intake would be associated with increased time in the recommended glucose range (TIR, 70–180 mg/dL) and reduced time below range (TBR, <70 mg/dL).

## 2. Materials and Methods

### 2.1. Parent Study

Secondary analyses were performed using data from the Metabolic, Hormonal, and Physiological Characterization of Isoenergetic High-Intensity Interval Training and Moderate Intensity Continuous Training in Adults with Type 1 Diabetes (HI1T) study (NCT04664205, PI Smith-Ryan). Review and approval of the HI1T study was provided by the institutional review board at the University of North Carolina at Chapel Hill (IRB Number: 20-3100). Written informed consent was provided by participants prior to participating in the HI1T study. The HI1T study enrolled 14 adults (7 male, 7 female) with T1D between the ages of 18 and 51 years old to participate in a randomized controlled exercise pilot study which aimed to characterize the metabolic, hormonal, and glycemic response to exercise among adults with T1D and explore the role of physiological variables (biological sex, lean body mass, visceral fat mass) in modulating the observed responses. 

Participants reported to the Applied Physiology Lab (APL) at the University of North Carolina at Chapel Hill for familiarization with study protocols and baseline testing, which consisted of body composition testing and a graded VO_2_ peak test. Following their baseline visit, participants reported to the APL for three additional measurement visits consisting of supervised HIIT, MICT, or a control visit (no exercise) in a randomized order and at least 1 week apart. Exercise sessions were performed on a cycle ergometer following an overnight fast. HIIT sessions consisted of 10 one-minute intervals at 90% VO_2_ peak followed by 1 min of active recovery. MICT sessions consisted of consistent work at 65% of VO_2_ peak for a time that allowed for equal energy expenditure as the HIIT sessions (typically 15–20 min). Blood draws were taken on arrival to the APL, immediately post-exercise and 1 h post-exercise or just on arrival and 1 h later for control sessions, which were then sent to the Duke Molecular and Physiology Institute Metabolomics Core for analysis of metabolic metabolites and metabolic hormone levels. Additionally, participants were asked to wear a continuous glucose monitor (CGM) and wearable physical activity tracker throughout the course of the study and also provide food records for the day before, day of, and day after each visit. 

### 2.2. Participants

Participants were recruited by phone or email from the UNC Endocrinology and Diabetes Clinic and UNC Student Health Center using data from the Carolina Data Warehouse to identify individuals with a type 1 diabetes diagnosis. Individuals with T1D between the ages of 18 and 51 years old with recent hemoglobin a1c (HbA1c) of <9%, a BMI < 30 kg/m^2^, a diabetes duration of at least one year, and who were otherwise healthy were eligible to participate in the HI1T study. Individuals with a physician diagnosis of active diabetic retinopathy, peripheral neuropathy with insensate feet, autonomic neuropathy, or a cardiovascular condition that would affect exercise tolerance, as well as those who were taking medications including beta-blockers, agents that affect hepatic glucose production, xanthine derivatives or any hypoglycemic agent other than insulin, were excluded from participating in the HI1T study. Additionally, individuals who had experienced severe hypoglycemia, defined as requiring a third party or hospitalization, diabetic ketoacidosis within the last 6 months, were pregnant, had severely impaired hearing or speech, were currently doing HIIT or those who used a closed-loop pump and were not willing to use manual mode were also excluded from participating in the HI1T study.

### 2.3. Measures

#### 2.3.1. Demographics and Health History

Demographics and health history questionnaires were completed at participants’ baseline visits, from which participants self-reported their age, sex, race/ethnicity, insurance, income, education, diabetes duration, and insulin regimen.

#### 2.3.2. Continuous Glucose Monitoring (CGM)

Participants were asked to either wear a study-provided intermittently scanned CGM system (Freestyle Libre Pro, Abbot Diabetes Care Inc., Alameda, CA, USA) for the duration of the study or share CGM data from their personal CGM devices for the time period in which they participated in the HI1T study. Participants with automated insulin delivery systems were required to utilize manual mode during the duration of this study. Percent time below range (TBR, <70 mg/dL), percent time in recommended glucose range (TIR, 70–180 mg/dL), and percent time above range (TAR, >180 mg/dL) were calculated from the cessation of exercise until 6:30 a.m. the following morning using raw CGM data exports following consensus guidelines [23]. As the effects of protein intake on glycemia have been shown to last at least 5 h following a meal [15,16,17], we chose to restrict observations to those with at least 5 h of CGM data following exercise. 

#### 2.3.3. Dietary Intake Measures

Participants were asked to complete three detailed food records of everything they ate and drank on the day before, the day of, and the day after each measurement visit. Food records were completed by participants using the ASA24 2020 automated food record system [24]. Email reminders were sent to participants on each day a food record was collected. From these records, total energy intake (kcals), as well as grams of protein, carbohydrate, and fat consumed from exercise cessation until midnight on the day of study visits, was quantified. As participants were asked to report to study visits following an overnight fast, participants had no dietary intake prior to or during exercise. 

#### 2.3.4. Physical Activity Measures

Participants were asked to wear a wearable activity tracker (Garmin vivosmart^®^ 4, Garmin Ltd., Olathe, KS, USA) throughout the duration of this study, which provided measures such as heart rate, step count, and minutes of moderate-to-vigorous physical activity (MVPA). 

#### 2.3.5. Anthropometrics and Body Composition

At their baseline visit, participants’ height and weight were measured using a wall-mounted stadiometer and calibrated electric scale. Body composition was also measured using a validated 4-compartment model utilizing dual-energy X-ray absorptiometry [25] (GE Lunar iDXA, GE Medical Systems Ultrasound & Primary Care Diagnostics, Madison, WI, USA) and bioelectrical impedance analysis (InBody570, BioSpace, Seoul, South Korea) which measured overall body fat percentage, body fat mass (kg), lean mass (kg), visceral fat mass (kg), bone mineral content (kg), and total body water. 

### 2.4. Statistical Analysis

#### 2.4.1. Model Selection

All statistical analyses were performed using SAS 9.4 (Cary, NC, USA). To account for repeated measures, mixed effects regression models were used to evaluate the proposed aims using the PROC MIXED command. Due to the small sample size for these analyses, only variables that were correlated with TAR, TIR, or TBR, as well as protein intake in grams or grams/kg at a *p*-value < 0.10, were considered for inclusion as potential covariates in our models. From these potential covariates, variables that caused a ≥10% change in the point estimate or standard error of associations were included in our final models. 

#### 2.4.2. Primary Analyses—Post-Exercise Protein Intake and Glycemia Following Exercise

An example timeline of exposures and outcomes relative to exercise sessions is provided in Figure 1. Post-exercise protein intake was defined continuously as absolute (g) or relative (g/kg bodyweight) protein intake from the cessation of exercise until midnight. Mixed effects regression models assessed the association between post-exercise protein intake and glycemia from the cessation of supervised bouts of exercise until 6:30 a.m. the following morning. Final models adjusted for diabetes duration (years), lean mass (kg), and carbohydrate intake (g). 

#### 2.4.3. Effect Measure Modification

Previous research has suggested that MICT and HIIT cause different post-exercise blood glucose trajectories with continuous aerobic exercise, such as MICT sessions, commonly being associated with declining glycemia, and high-intensity exercise, such as HIIT or resistance exercise, commonly causing glycemia to increase during and following exercise [26,27,28]. It is possible that differing trajectories in blood glucose due to the type of exercise session may influence the association between protein intake and post-exercise glycemia. As such, we explored potential effect measure modification by stratifying our final models by the type of exercise session performed. Potential differences in absolute and relative protein intake, as well as TAR, TIR, and TBR between HIIT and MICT sessions, were assessed utilizing unadjusted mixed effects models. Due to the low sample size of this study, we also chose to graph individual responses to post-exercise on TBR by exercise modality to confirm the trends observed in primary statistical analyses and to assess for possible outlier effects.

## 3. Results

### 3.1. Final Sample Size

Fourteen participants (seven male and seven female) were recruited to participate in the HI1T study. As each participant reported to the Applied Physiology Lab for two exercise visits as part of the study, we had 28 initial observations. Of these twenty-eight observations, five observations from two participants had missing (two observations from one participant) or insufficient (three observations from two participants) CGM data. An additional two observations from two participants were missing dietary intake data, and two observations from one participant were missing a date of diabetes diagnosis from which to determine their duration of diabetes. Our final sample size for these analyses included 19 observations from 11 participants. 

### 3.2. Baseline CharacteristicsTh

Baseline characteristics for the participants included in these analyses are provided in Table 1. Participants were between the ages of 21 and 50 years of age with an approximately equal distribution of male and female participants and an average BMI, hemoglobin a1c (HbA1c), and diabetes duration of 25.1 ± 3.4 kg/m^2^, 6.5 ± 0.8%, and 17.0 ± 13.5 years. Additionally, all participants reported using a CGM at least 10 days in the past 30 days, and there was a similar number of individuals utilizing continuous subcutaneous insulin infusion (54.6%) or multiple daily injects (45.4%) for their insulin regimen. Of the eleven participants included in our final analyses, one (9%) used a study provided Freestyle LibrePro, ten used their personal CGM, of which six (55%) used a Dexcom CGM, one (9%) used a Medtronic CGM, and three used a personal Freestyle LibrePro (27%).

### 3.3. Results of Primary Analyses

The median intake of dietary protein following bouts of exercise until midnight was 75.0 (IQR: 57.5, 98.8) grams or 0.94 (IQR: 0.76, 1.17) grams/kilogram of body weight. Median TAR, TIR, and TBR following exercise sessions were 21.0% (IQR: 12.5%, 52.1%), 73.6% (IQR: 46.2%, 87.3%), and 1.3% (IQR: 0.0%, 4.1%), respectively. Results for primary analyses assessing the relationship between absolute (grams) and relative (grams/kilogram bodyweight) post-exercise protein intake on glycemia following exercise are provided in Table 2. While there appeared to be a trend towards lower TBR following exercise for both absolute protein intake, −1.2% (95% CI: −2.6%, 0.3%) per 20 g protein (*p* = 0.09), and relative protein intake, −1.0% (95%CI: −2.1%, 0.1%) per 0.25 g/kg protein (*p* = 0.07), these results were not statistically significant. No statistically significant associations were observed between absolute protein intake with TIR, −1.7% (95% CI: −9.3%, 5.8%) per 20 g protein (*p* = 0.60), or TAR, 2.9% (95% CI: −5.1%, 10.9%) per 20g protein (*p* = 0.41). Similarly, no association was observed between relative protein intake and TIR, −1.5% (95% CI: −7.4%, 4.4%) per 0.25 g/kg protein (*p* = 0.55), or TAR, 2.3% (95% CI: −4.0%, 8.6%) per 0.25 g/kg protein (*p* = 0.41).

### 3.4. Results of Effect Measure Modification

No significant differences were observed between HIIT and MICT sessions for any of the exposures or outcome variables (*p*-values > 0.36). The results of stratified models are provided in Table 3. In stratifying final models by exercise session type, it was observed that absolute protein intake (grams) was borderline associated with reduced TBR following bouts of MICT, −1.9% (95% CI: −3.9%, 0.0%) per 20 g protein (*p* = 0.05), but not following HIIT sessions, 1.2% (95% CI: −2.4%, 4.9%) per 20 g protein (*p* = 0.42). Relative protein intake (g/kg) was not associated with TBR following MICT, −1.2% (95% CI: −3.0%, 0.6%) per 0.25 g/kg (*p* = 0.14), or HIIT sessions, 2.4% (95% CI: −1.0%, 5.8%) per 0.25 g/kg (*p* = 0.13). No significant associations were observed between absolute protein intake and TIR or TAR for MICT, −3.0% (95%CI: −21.2%, 15.2%) TIR and 4.9% (95% CI: −13.5% 23.4%) TAR per 20 g protein, or HIIT, 5.3% (95% CI: −8.0%, 18.7%) TIR and −6.6% (95% CI: −18.3%, 5.25%) TAR per 20 g protein. Similarly, no statistically significant associations were observed between relative protein intake and TIR or TAR for MICT, −1.1% (95% CI: −14.8%, 12.7%) TIR and 2.3% (95% CI: −12.0%, 16.5%) TAR per 0.25 g/kg protein, or HIIT, −0.9% (95%CI: −17.3%, 15.5%) TIR and 2.3% (95% CI: −17.0%, 14.0%) TAR per 0.25 g/kg protein. Individual responses to post-exercise protein intake on post-exercise TAR, TIR, and TBR are illustrated in Appendix A.

## 4. Discussion

This study evaluated the relationship between free-living post-exercise protein intake and glycemia following isoenergetic bouts of MICT or HIIT among 11 adults with T1D. We hypothesized that higher post-exercise protein intake would be associated with improved TIR and reduced TBR following exercise sessions until the following morning. Overall, no significant associations were observed between post-exercise protein intake and post-exercise TIR, TBR, or TAR. However, there was a trend towards reduced TBR with increasing protein intake (*p* < 0.1). In stratified analyses, we observed a borderline significant association suggesting increasing absolute protein intake (grams), but not relative protein intake (grams), may be associated with reduced TBR following MICT sessions (*p* = 0.05) but not HIIT sessions (*p* = 0.42). No significant associations were observed between post-exercise protein intake and TIR or TAR for MICT or HIIT sessions in stratified analyses. 

When examining individual responses to post-exercise protein intake and post-exercise TBR (Appendix A), the pattern of responses appears to be in agreement with mixed effects regression results, suggesting a trend towards reduced TBR following MICT with increasing post-exercise protein intake, and does not appear to be affected by outlying observations. Another important takeaway from these results is the range of times spent in hypoglycemia by participants. Nearly one in four observations demonstrated TBR greater than clinical guidelines, which recommend the minimization of glycemic excursions with a target of < 4% TBR. Additionally, several observations had >8% TBR, or ~105 min TBR, highlighting the clinical significance of post-exercise glycemia in this population [29].

While there did not appear to be a trend in individual responses of increasing post-exercise protein intake on TIR following either HIIT or MICT (Appendix A), there appeared to be a potential trend towards reduced TAR with increasing post-exercise protein intake following HIIT (Appendix A), but not MICT (Appendix A), that may have been affected by an outlying observation. After rerunning stratified analyses following the removal of this potential outlier, however, associations for relative (−5.8%, *p* = 0.3) and absolute protein intakes (−6.6%, *p* = 0.11) on TAR following HIIT remained statistically non-significant.

The findings of this study suggest that elevating protein intake following moderate-intensity continuous training, as is recommended by sports nutrition guidelines [8], may reduce the risk of experiencing post-exercise hypoglycemia following aerobic exercise sessions until the following morning. A 20 g dose of exercise protein intake, approximately the amount provided by 3 oz of chicken or one scoop of a protein supplement, was associated with a −1.9% (95% CI: −3.9%, 0.0%) reduction in TBR. Given that the median TBR following exercise sessions was 1.3% (IQR: 0.0%, 4.1%), which is equivalent to ~15.5 min, a reduction of 1.9% or ~24.5 min is clinically significant. While no association was observed between post-exercise protein intake and glycemia following bouts of HIIT, this may be explained by differences in the glycemic response during and following exercise between these different exercise modalities. While results have been inconsistent, a meta-analysis of exercise trials in adults with T1D suggested that intermittent high-intensity exercise, such as HIIT, may cause less of a decline in blood glucose levels following exercise compared to continuous moderate-intensity exercise, possibly due to an increased counter-regulatory response to higher intensity activity [27]. Additionally, the prandial state in which HIIT exercise appears to modulate the glycemic effect observed following exercise among people with T1D [30,31,32]. Recent studies have observed that when HIIT is performed in a fasted state, as in this study, blood glucose levels tend to increase following exercise, whereas they tend to decrease following HIIT exercise performed in a post-prandial state [30,31,32]. 

While research investigating the effects of post-exercise protein intake on glycemia among individuals with T1D is scarce, a study by Paramalingam et al. found that a 50 g bolus of protein following moderate-intensity continuous exercise reduced the glucose requirement to maintain euglycemia overnight compared to water [22]. This effect was attributed in part to an increase in glucagon, which may stimulate hepatic glucose production [22]. Another study among adolescents with T1D (*n* = 10) observed that a protein-supplemented breakfast consumed two hours prior to exercise raised blood glucose levels at the onset of exercise compared to a standard breakfast, leading to a reduced occurrence of hypoglycemia during exercise [21]. While speculative, it is possible that increasing hepatic glucose production, possibly through gluconeogenesis utilizing specific amino acids from high protein meals, may help to mitigate the decline in glycemia observed following exercise. More research is needed to determine whether a causal relationship exists between post-exercise protein intake and the risk of hypoglycemia following exercise and to provide insights into potential mechanisms by which post-exercise protein intake effects the post-exercise glucose response. 

### 4.1. Challenges and Opportunities

A primary limitation of this study is that, as it uses data from a pilot study, there was a limited sample size to evaluate the aims of the study (*n* = 11, obs = 19). In addition to limiting the statistical power of our analyses, the small sample size restricted the number of potential covariates that could be incorporated into our statistical models, thus increasing the potential for residual confounding. Additionally, while the findings of this study provide insights that may inform future studies, the observational nature of our analyses limits our ability to evaluate a causal relationship between post-exercise protein intake and post-exercise glycemia. As individuals in this study had very good glycemic control at baseline (mean HbA1c 6.5 ± 0.8%) and reported lower levels of kcals from protein compared to other studies in adults with T1D (T1D Exchange: 18.2 ± 0.6% vs. HI1T: 14.9 ± 4.7%), there is a potential that our results may be influenced by potential selection bias which may bias our results away from the mean [33]. Furthermore, self-reported measures of dietary intake are prone to under-reporting [34]; however, the extent of mis-reporting has been shown to be reduced when the multiple pass method is utilized, as in the ASA-24 system that was utilized in this study [35]. Additionally, while carbohydrate intake was controlled, which may help to account for bolus insulin provision, which is based on carbohydrate intake, the lack of insulin dosing data limits our ability to assess whether changes in insulin-dosing behaviors may have influenced the observed results. Finally, since participants in this study were allowed to either use a study-provided CGM or share the data from their own personal CGM, there may be bias due to differences in the error of measurement of interstitial blood glucose by CGM type. 

This study also had several strengths. One strength of this study was the provision of supervised, iso-energetic bouts of HIIT and MICT exercise sessions, which reduces the risk of confounding related to differences in energy expenditure between bouts of exercise. Additionally, the use of continuous glucose monitoring throughout the study provided the opportunity to examine glycemia over a fuller extent of the time period for which the risk of hypoglycemia is elevated following exercise. Additionally, while observational in nature, these data allowed us to address an important gap in the literature relating to the role of post-exercise protein intake on post-exercise glycemia. 

### 4.2. Relevance for Clinical Practice

Increasing protein intake following exercise is recommended by sports nutrition guidelines for promoting numerous physiological adaptations to exercise training [8]. Following aerobic exercise, increasing protein intake has been associated with reduced muscle damage and soreness and improved recovery following exercise [14,36,37,38]. Additionally, following resistance exercise, elevating protein intake has been associated with improvements in muscular strength and hypertrophy [11,39], and when paired with a caloric deficit, high protein diets may promote reductions in fat mass and retention of lean mass during weight loss [9,40]. While these findings have been observed largely in healthy populations, it is likely people with T1D may experience similar adaptive benefits to increasing protein intake post-exercise. Our results suggest that, among people with T1D, following this same nutritional strategy may potentially reduce the risk of post-exercise hypoglycemia, specifically following MICT exercise. Our results also suggested that increasing post-exercise protein intake is not associated with any detrimental changes in glycemia following either MICT or HIIT sessions. While more research is needed, increasing protein intake post-exercise may pose a promising strategy for both mitigating the risk of hypoglycemia following exercise and improving the adaptive response to exercise among people living with T1D. These preliminary data will be valuable in informing future clinical trials, which will provide more specific evidence of the potential benefits and adequate dosing of protein intake in maintaining euglycemia following exercise among adults with T1D.

Additionally, the development of smart algorithms to provide carbohydrate intake and insulin dosing adjustments to help people with T1D maintain euglycemia has demonstrated promising results [41,42,43]. Incorporating dietary protein intake into post-exercise nutrition recommendations with smart algorithms may be beneficial following more moderate-intensity continuous exercise as an additional strategy to help prevent post-exercise hypoglycemia; however, caution should be taken in incorporating dietary protein intake into insulin dose adjustments recommendations following exercise. While studies have shown that high protein intake may require insulin to maintain euglycemia [44,45,46], these studies were not conducted in the context of exercise when insulin sensitivity is heightened. Clinical trials are needed to determine whether insulin dose adjustments for protein intake following exercise are necessary or beneficial, as correcting insulin doses for protein in this context may hamper the potential benefit of dietary protein in preventing post-exercise hypoglycemia.

### 4.3. Future Directions

Research in the field of peri-exercise protein intake and exercise-related glycemia among people with T1D is very limited but warrants further study. Specifically, randomized controlled studies are needed to establish whether a causal relationship exists between post-exercise protein intake and post-exercise glycemia among adults with T1D. As fear of hypoglycemia is a leading barrier to regular participation in physical activity for people with T1D, identification of nutritional strategies that may help to address this fear while also providing other potential health and performance benefits would have significant implications for clinical practice. While current strategies around carbohydrate consumption are effective in treating or preventing hypoglycemia [7], young adults with T1D have reported that the need to consume carbohydrates to prevent hypoglycemia with exercise can create a feeling of futility around exercise when weight management is a primary goal for exercise [47]. Future work should strive to bridge the fields of sports nutrition and diabetes care to identify nutritional strategies that may aid adults in exercising safely while also supporting the goals that have motivated them to engage in more regular exercise. 

## Figures and Tables

**Figure 1 nutrients-15-04268-f001:**
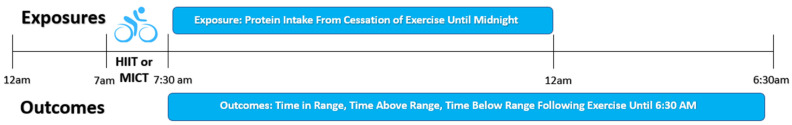
Example timeline of exposures and outcomes relative to exercise sessions. All exercise sessions were performed following an overnight fast.

**Table 1 nutrients-15-04268-t001:** Baseline characteristics of HI1T participants included in final analyses (*n* = 11).

Demographic	Mean ± SD or N (%)
Age	33 ± 11.4
Female	6 (54.6)
Male	5 (45.5)
Self-Reported Race/Ethnicity	
Non-Hispanic White	8 (72.7)
Asian	2 (18.2)
Hispanic	1 (9.1)
Diabetes Care	
Diabetes Duration (Years)	17.0 ± 13.5
Insulin Regimen	
Continuous Subcutaneous Insulin Infusion	6 (54.6)
Multiple Daily Injections	5 (45.4)
≥10 Days CGM Use in Past 30 Days	11 (100%)
Most Recent HbA1c (%) (*n* = 10)	6.5 ± 0.8
Anthropometric	
Weight (kg)	73.9 ± 13.4
BMI	25.1 ± 3.4
Estimated Body Fat %	26.8 ± 7.7
Total Lean Mass (kg)	51.0 ± 10.5
Total Body Fat Mass (kg)	19.0 ± 7.5
Visceral Fat Mass (kg)	0.4 ± 0.3
Diet	
Daily Caloric Intake (kcal)	1675.2 (1417.8, 2040.6)
Percent of Daily Calories from Protein	14.9 ± 4.7
Percent of Daily Calories from Carbohydrate	40.2 ± 13.6
Percent of Daily Calories from Fat	42.5 ± 9.5
Daily Fiber Intake (Grams)	11.4 (10.5, 15.6)
VO_2_ Peak (mL/kg/min)	31.1 ± 9.1

Continuous variables are reported as mean and standard deviation except for non-normally distributed variables, in which median and interquartile range are reported. Categorical variables are described with counts and percentages.

**Table 2 nutrients-15-04268-t002:** Overal results of mixed effects regression models assessing the association between post-exercise protein intake and glycemia following supervised exercise (*n* = 11, obs = 19).

	Post-Exercise Protein (grams) *			Post-Exercise Protein (g/kg) †		
	Estimate	*p*-Value	95% CI	Estimate	*p*-Value	95% CI
	Unadjusted Models					
Percent Time Above Range	2.7%	0.48	(−5.8%, 11.2%)	0.8%	0.76	(−5.2%, 6.9%)
Percent Time In Range	−1.7%	0.66	(−10.1%, 6.7%)	−0.1%	0.96	(−6.1%, 5.8)
Percent Time Below Range	−0.8%	0.10	(−1.9%, 0.2%)	−0.6%	0.10	(−1.3%, 0.1%)
	Fully Adjusted Models ‡					
Percent Time Above Range	2.9%	0.41	(−5.1%, 10.9%)	1.8%	0.41	(−4.0%, 8.6%)
Percent Time In Range	−1.7%	0.60	(−9.3%, 5.8%)	−1.2%	0.55	(−7.4%, 4.4%)
Percent Time Below Range	−1.2%	0.09	(−2.6%, 0.3%)	−1.0%	0.07	(−2.1%, 0.1%)

* Associations are reported per a 20 g dose of protein; † associations are reported per a 0.25 g/kg dose of protein; ‡ models are adjusted for diabetes duration, lean mass (kg), and carbohydrate intake.

**Table 3 nutrients-15-04268-t003:** Results of mixed effects regression models assessing the association between post-exercise protein intake and glycemia following supervised exercise by exercise modality (*n* = 11, obs = 19).

	Post-Exercise Protein (grams) *			Post-Exercise Protein (g/kg) †		
	Estimate	*p*-Value	95% CI	Estimate	*p*-Value	95% CI
Moderate Intensity Continuous Training (MICT)						
Unadjusted Models (*n* = 10, obs = 10)						
Time Above Range	−1.30%	0.85	(−17.1%, 14.4%)	−3.10%	0.53	(−14.1%, 7.8%)
Time In Range	3.00%	0.66	(−12.4%, 18.4%)	4.10%	0.39	(−6.4%, 14.6%)
Time Below Range	−1.70%	0.04	(−3.3%, −0.1%)	−1.00%	0.11	(−2.2%, 0.3%)
Fully Adjusted Models ^‡^ (*n* = 9, obs = 9)						
Time Above Range	4.90%	0.5	(−13.5%, 23.4%)	2.30%	0.46	(−12.0%, 16.5%)
Time In Range	−3.00%	0.67	(−21.2%, 15.2%)	−1.10%	0.84	(−14.8%, 12.7%)
Time Below Range	−1.90%	0.05	(−3.9%, 0.0)	−1.20%	0.14	(−3.0%, 0.6%)
High-Intensity Interval Training (HIIT)						
Unadjusted Models (*n* = 11, obs = 11)						
Time Above Range	5.00%	0.27	(−4.62%, 14.6%)	6.20%	0.21	(−4.3%, 16.8%)
Time In Range	−4.50%	0.29	(−13.8%, 4.7%)	−6.40%	0.18	(−16.3%, 3.5%)
Time Below Range	−0.50%	0.53	(−2.0%, 1.1%)	0.10%	0.86	(−1.7%, 1.9%)
Fully Adjusted Models ^‡^ (*n* = 10, obs = 10)						
Time Above Range	−6.60%	0.21	(−18.3%, 5.2%)	−1.50%	0.82	(−17.0%, 14.0%)
Time In Range	5.30%	0.35	(−8.0%, 18.7%)	−0.90%	0.89	(−17.3%, 15.5%)
Time Below Range	1.20%	0.42	(−2.4%, 4.9%)	2.40%	0.13	(−1.0%, 5.8%)

* Associations are reported per a 20 g dose of protein; † associations are reported per a 0.25 g/kg dose of protein; ‡ models are adjusted for diabetes duration, lean mass (kg), and carbohydrate intake.

## Data Availability

Deidentified individual data that supports the results will be shared beginning 9 to 36 months following publication provided the investigator who proposes to use the data has a specific research question and approval from an Institutional Review Board (IRB), Independent Ethics Committee (IEC), or Research Ethics Board (REB), as applicable, and executes a data use/sharing agreement with UNC.

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
