# Peer review of "Post-Exercise Protein Intake May Reduce Time in Hypoglycemia Following Moderate-Intensity Continuous Exercise among Adults with Type 1 Diabetes"

_nutrients, 2023, doi:10.3390/nu15194268_

Round 1
Reviewer 1 Report
This study analyzes important preventive measures against postprandial hypoglycemia in type 1 diabetes.
However, an important issue needs to be considered.
This study is not the best study design for the research goal of assessing the potential of post-exercise protein intake to reduce the risk of postexercise late-onset hyperglycemia.
This study is being conducted from a secondary analysis to one pilot study, but in the end, it would be faster to collect new type 1 diabetic patients if 11 were to be analyzed.
Also, 6 patients with CSII were included, but it was not stated whether these patients were using AIDs. evaluation of TBR under rtCGm use is largely dependent on patient awareness.
Reviewer 2 Report
I would like to thank you for the opportunity to review this excellent manuscript providing preliminary evidence about the potential use of protein intake after exercise to reduce the risk of glycemic imbalances in people with T1D. Promoting active lifestyles and safe physical activity is mandatory to improve and maintain health in people with T1D, as for many reasons, the clinical conditions still represent a barrier to exercise in many individuals.
The study has been well-designed and conducted, and the manuscript is clearly written. The topic is of interest and despite the sample is moderate, the conclusions are supported by the results. I really do not have any additional comment or suggestion to improve this work.
Again, congratulations for the interesting and well-done study.
My only curiosity relates to the discussion about the problems in controlling post-exercise glycemia due to the different CHO intake. Some studies have provided evidence that smart solutions and predictive models might help people with T1D to adapt their CHO intake according to the exercise, diet and therapy characteristics (Buoite Stella et al., 2017; Kilbride et al., J Clin Nurs, 2011; Lysy et al., Front Endocrinol, 2021). Do the authors think that the the possible supportive role of post-exercise protein should be implemented and taken into account in such algorithms? Maybe, it could be used as an additional strategy to maintain euglycemia, and if people aim to consume proteins after the exercise, these algorithms should consider it when planning the suggested CHO/insulin adjustments.
Round 2
Reviewer 1 Report
If it is a pilot study, it would be better to mention it in the title.
I missed a sentence about AID. My apologies.
Am I correct in understanding that PLGM is not used for this either?